# How Robust is Your Fairness? Evaluating and Sustaining Fairness under Unseen Distribution Shifts

**Haotao Wang**                                                                          *htwang@utexas.edu*
*University of Texas at Austin*

**Junyuan Hong**                                                                         *hongju12@msu.edu*
*Michigan State University*

**Jiayu Zhou**                                                                           *jiayuz@msu.edu*
*Michigan State University*

**Zhangyang Wang**                                                                       *atlaswang@utexas.edu*
*University of Texas at Austin*

**Reviewed on OpenReview:** *https://openreview.net/forum?id=11pGlecTz2*

## Abstract

Increasing concerns have been raised on deep learning fairness in recent years. Existing fairness-aware machine learning methods mainly focus on the fairness of in-distribution data. However, in real-world applications, it is common to have distribution shift between the training and test data. In this paper, we first show that the fairness achieved by existing methods can be easily broken by slight distribution shifts. To solve this problem, we propose a novel fairness learning method termed CUrvature MAtching (CUMA), which can achieve robust fairness generalizable to unseen domains with unknown distributional shifts. Specifically, CUMA enforces the model to have similar generalization ability on the majority and minority groups, by matching the loss curvature distributions of the two groups. We evaluate our method on three popular fairness datasets. Compared with existing methods, CUMA achieves superior fairness under unseen distribution shifts, without sacrificing either the overall accuracy or the in-distribution fairness.

## 1 Introduction

With the wide deployment of deep learning in modern business applications concerning individual lives and privacy, there naturally emerge concerns on machine learning fairness (Podesta et al., 2014; Muñoz et al., 2016; Smuha, 2019). Although research efforts on various fairness-aware learning algorithms have been carried out (Edwards & Storkey, 2016; Hardt et al., 2016; Du et al., 2020), most of them focus only on equalizing model performance across different groups on *in-distribution* data.

Unfortunately, in real-world applications, one commonly encounters data with unforeseeable distribution shifts from model training. It has been shown that deep learning models have drastically degraded performance (Hendrycks & Dietterich, 2019; Hendrycks et al., 2020; 2021; Taori et al., 2020) and show unpredictable behaviors (Qiu et al., 2019; Yan et al., 2021) under unseen distribution shifts. Most previous fairness learning algorithms aim to achieve fairness on in-distribution data. However, those algorithms do not take into consideration the stability or "robustness" of their found fairness-aware minima. Taking object detection in self-driving cars for example, it might have been calibrated over high-quality clear images to be "fair" with different pedestrian skin colors; however such fairness may break down when applied to data collected in adverse visual conditions, such as inclement weather, poor lighting, or other digital artifacts. Our experiments also find that previous state-of-the-art fairness algorithms would be jeopardized if distributional shifts are present in test data. The above findings beg the following question:

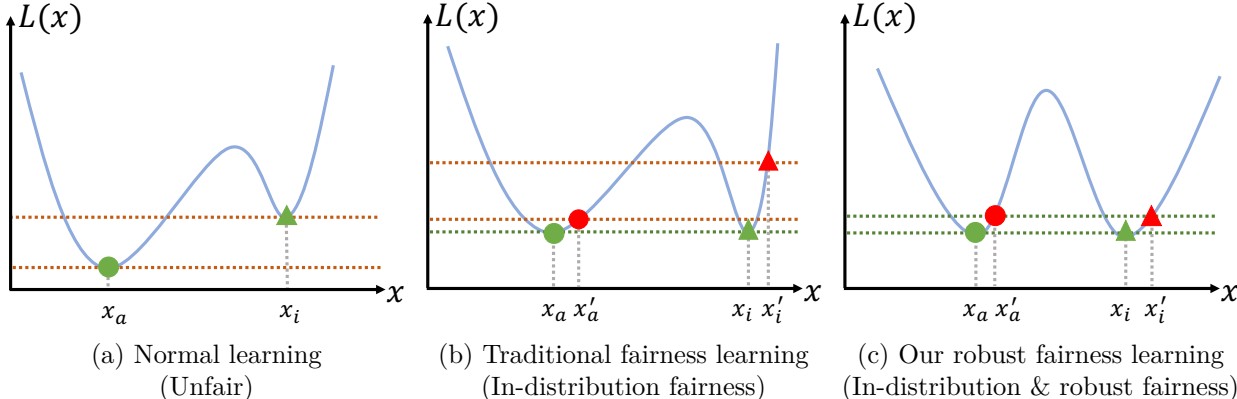

(a) Normal learning
(Unfair)

(b) Traditional fairness learning
(In-distribution fairness)

(c) Our robust fairness learning
(In-distribution & robust fairness)

Figure 1: Illustrating the achieved fairness of normal training, traditional fair training and our proposed robust fair training algorithms. Horizontal and vertical axes represent input $x$ and corresponding loss value $\mathcal{L}(x)$, respectively. Solid blue curves show the loss landscapes. Circles denote majority data points ($x_a$ and $x_a'$), while triangles denote minority data points ($x_i$ and $x_i'$). Green points ($x_a$ and $x_i$) are in-distribution data while red ones ($x_a'$ and $x_i'$) are sampled from test sets with distribution shifts. (a) Normal training results in unfair models: minority group has worse performance (i.e., larger loss values). (b) Traditional fairness learning algorithms can achieve in-distribution fairness but not in a robust way: a small distribution shift can break the fairness due to loss curvature biases across different groups. In fact, such learned fair models can have almost the same large bias as the normally trained models when facing distribution shifts. (c) Our robust fairness learning algorithm can simultaneously achieve fairness both on in-distribution data and at distribution shifts, by matching both loss values and loss curvatures across different groups.

*Using the currently available training set, how to achieve robust fairness that is generalizable to unseen domains with unpredictable distribution shifts?*

To solve this problem, we decompose it into the following two-step objectives and achieve them one by one:

1. The minority and majority groups should have similar prediction **true positive rates** and **false positive rates** on *in-distribution* data. This is usually attained by traditional in-distribution fairness goals.

2. The minority and majority groups should have similar **robustness** under *unseen distributional shifts*. In this context, the "robustness" refers to model performance gap between in-distribution and unseen out-of-distribution data: the larger gap the weaker.

The first objective is well studied and can be achieved by existing fairness learning methods such as adversarial training Edwards & Storkey (2016); Hardt et al. (2016); Du et al. (2020). In this paper, we focus our efforts on addressing the second objective, which has been much less studied. We present empirical evidence that the fairness achieved by existing in-distribution oriented methods can be easily compromised by even slight distribution shifts.

Next, to mitigate this fragility, we note that model robustness against distributional shift is often found to be highly correlated with the loss curvature (Bartlett et al., 2017; Weng et al., 2018). Our experiments further observed that, the local loss curvature of minority group is often much larger than that of majority group, which explains their discrepancy of robustness. Motivated by the above, we propose a new fairness learning algorithm, termed **Curvature Matching (CUMA)**, to robustify the fairness. Specifically, CUMA enforces the model to have similar robustness on the majority and minority groups, by matching the loss curvature distributions of the two groups. As a plug-and-play modular, CUMA can be flexibly combined with existing in-distribution fairness learning methods, such as adversarial training, to fulfil our overall goal of robust fairness. We illustrate the core idea of CUMA and its difference compared with traditional in-distribution fairness methods in Figure 1.We evaluate our method on three popular fairness datasets: Adult, C&C, and CelebA. Experimental results show that CUMA achieves significantly more robust fairness against unseen

distribution shifts, without sacrificing either overall accuracy or the in-distribution fairness, compared to traditional fairness learning methods.

## 2 Preliminaries

### 2.1 Machine Learning Fairness

**Wide Existence of Unfairness in Machine Learning**   With the wide application of machine learning in real-world applications, increasing concerns have been raised on the potential unfairness in machine learning model and algorithms. For instance, Buolamwini & Gebru (2018) shows current commercial gender-recognition systems have substantial accuracy disparities among groups with different genders and skin colors. Wilson et al. (2019) observe that state-of-the-art segmentation models achieve better performance on pedestrians with lighter skin colors. In (Shankar et al., 2017; de Vries et al., 2019), it is found that the common geographical bias in public image databases can lead to strong performance disparities among images from locales with different income levels. Nagpal et al. (2019) reveal that the focus region of face-classification models depends on people's ages or races, which may explain the source of age- and race-biases of classifiers. On the awareness of the unfairness, many efforts have been devoted to mitigate such biases in computer vision tasks.

**Problem Setting and Metrics**   Machine learning fairness can be generally categorized into individual fairness and group fairness (Du et al., 2020). Individual fairness requires similar inputs to have similar predictions (Dwork et al., 2012). This paper focuses on group fairness, whose definition is provided below. Given input data $X \in \mathbb{R}^n$ with sensitive attributes $A \in \{0, 1\}$ and their corresponding ground truth labels $Y \in \{0, 1\}$, group fairness requires a learned binary classifier $f(\cdot; \theta) : \mathbb{R}^n \to \{0, 1\}$ parameterized by $\theta$ to give equally accurate predictions (denoted as $\hat{Y} := f(X)$) on the two groups with $A = 0$ and $A = 1$. Multiple fairness criteria have been defined in this context. Demographic parity (DP) (Edwards & Storkey, 2016) requires identical ratio of positive predictions between two groups: $P(\hat{Y} = 1|A = 0) = P(\hat{Y} = 1|A = 1)$. Equalized Odds (EO) (Hardt et al., 2016) requires identical false positive rates (FPRs) and false negative rates (FNRs) between the two groups: $P(\hat{Y} \neq Y|A = 0, Y = y) = P(\hat{Y} \neq Y|A = 1, Y = y), \forall y \in \{0, 1\}$. Based on these fairness criteria, quantified metrics are defined to measure fairness. For example, the EO and EOpp distances (Madras et al., 2018) are defined as follows:

$$\Delta_{EO} := \sum_{y \in \{0,1\}} |P(\hat{Y} \neq Y|A = 0, Y = y) - P(\hat{Y} \neq Y|A = 1, Y = y)|, \tag{1}$$

$$\Delta_{EOpp} := |P(\hat{Y} \neq Y|A = 0, Y = 1) - P(\hat{Y} \neq Y|A = 1, Y = 1)|. \tag{2}$$

Mary et al. (2019) proposed to generalize the fairness measurements from binary variables to multi-variate or continuous variables by using the Renyi maximum correlation coefficient.

**Bias Mitigation Methods**   Many methods have been proposed to mitigate model bias. Data pre-processing methods such as re-weighting (Kamiran & Calders, 2012) and data-transformation (Calmon et al., 2017) have been used to reduce discrimination before model training. In contrast, Hardt et al. (2016) and Zhao et al. (2017) propose post-processing methods to calibrate model predictions towards a desired fair distribution after model training. Instead of pre- or post-processing, researchers have explored to enhance fairness during training. For example, Madras et al. (2018) uses a adversarial training technique and shows the learned fair representations can transfer to unseen target tasks. The key technique, adversarial training (Edwards & Storkey, 2016), was designed for feature disentanglement on hidden representations such that sensitive (Edwards & Storkey, 2016) or domain-specific information (Ganin et al., 2016) will be removed while keeping other useful information for the target task. The hidden representations are typically the output of intermediate layers of neural networks (Ganin et al., 2016; Edwards & Storkey, 2016; Madras et al., 2018). Instead, methods, like adversarial debiasing (Zhang et al., 2018) and its simplified version (Wadsworth et al., 2018), directly apply the adversary on the output layer of the classifier, which also promotes the model fairness. Observing the unfairness due to ignoring the worst learning risk of specific samples, Hashimoto et al. (2018) proposes to use distributionally robust optimization which provably bounds the worst-case risk over groups. Creager et al. (2019) proposes a flexible fair representation learning framework based on VAE

(Kingma & Welling, 2013), that can be easily adapted for different sensitive attribute settings during run-time. Sarhan et al. (2020) uses orthogonality constraints as a proxy for independence to disentangles the utility and sensitive representations. Martinez et al. (2020) formulates group fairness with multiple sensitive attributes as a multi-objective learning problem and proposes a simple optimization algorithm to find the Pareto optimality. Another line of research focuses on learning unbiased representations from biased ones (Bahng et al., 2020; Nam et al., 2020). Bahng et al. (2020) proposes a novel framework to learn unbiased representations by explicitly enforcing them to be different from a set of pre-defined biased representations. Nam et al. (2020) observes that data bias can be either benign or malicious, and removing malicious bias along can achieve fairness. Li & Vasconcelos (2019) jointly learns a data re-sampling weight distribution that penalizes easy samples and network parameters. Li et al. (2019) scaled by higher-order power to re-emphasize the loss of minority samples (or nodes) in distributed learning. Agarwal et al. (2018) formulates a fairness-constrained optimization to train a randomized classifier which is provably accurate and fair. Quadrianto et al. (2019) casts the sensitive information removal problem as a data-to-data translation problem with unknown target domain. Wang et al. (2019) shows the effectiveness of adversarial debiasing technique (Zhang et al., 2018) in fair image classification and activity recognition tasks. Beyond the supervised learning, FairFaceGAN (Hwang et al., 2020) is proposed to prevent undesired sensitive feature translation during image editing. Similar ideas have also been successfully applied to visual question answering (Park et al., 2020). Szabó et al. (2021) proposed tilted cross-entropy to achieve fairness in semantic segmentation. Prost et al. (2019) proposed to improve the trade-off between performance and fairness by kernel-based distribution matching. Baharlouei et al. (2019) used Renyi correlation as a regularization term in training to impose group fairness. Cho et al. (2020) used kernel density estimation to quantify fairness measurements in a differentiable way, which is used in training to optimize the trade-off between accuracy and fairness. Recently, Lowy et al. (2022) developed the first stochastic in-processing fairness algorithm with guaranteed convergence in stochastic optimization with any batch size. The proposed method, named FERMI, is capable of achieving fairness on multiple (non-binary) sensitive attributes and non-binary targets, performing well even with minibatch size as small as one. FERMI empirically achieves state-of-the-art performance across multiple different problem settings, outperforming previous methods by a significant margin. For more related works on machine learning fairness, please refer to the recent survey (Hort et al., 2022).

**Fairness under distributional shift** Recently, several papers have investigated the fairness learning problem under distributional shift (Mandal et al., 2020; Zhang et al., 2021; Rezaei et al., 2021; Singh et al., 2021; Dai & Brown, 2020). Although these works are relevant with ours, there are significant differences in the problem settings. Zhang et al. (2021) studied the problem of enforcing fairness in online learning, where the training distribution constantly shifts. The authors proposed to adapt the model to be fair on the current *known* data distribution. In contrast, our work aims to generalize fairness learned on current distribution to *unknown* and *unseen* target distributions. In our setting, the algorithm can not access any training data from the unknown target distributions. Rezaei et al. (2021) studied to preserve fairness under covariate shift. However, their method requires unlabeled data from the target distribution. In other words, they assume the target distribution to be *known*. In contrast, our method is more general and works on *unknown* target distributions. Singh et al. (2021) also studied to preserve fairness under covariate shift. However, their method is based on model adaptation and requires the existence of a joint causal graph to represent the data distribution for all domains. Our method, however, does not require such and generally works on any unseen target distributions. Dai & Brown (2020) studies fairness under label distributional shift, while we focus on covariate shift and temporal shift. Mehrabi et al. (2021) proposed data poisoning attacks targeting fairness.

## 2.2 Model Robustness and Smoothness

Model generalization and robustness have been shown to be highly correlated with model smoothness (Moosavi-Dezfooli et al., 2019; Weng et al., 2018). Weng et al. (2018) and Guo et al. (2018) use the local Lipschitz constant to estimate model robustness against small perturbations on inputs within a hyper-ball. Moosavi-Dezfooli et al. (2019) proposes to improve model robustness by adding a curvature constraint to encourage model smoothness. Miyato et al. (2018) approximates model local smoothness by the spectral norm of Hessian matrix, and improves model robustness against adversarial attacks by regularizing model smoothness.

## 3    The Challenge of Robust Fairness

In this section, we show that the current state-of-the-art in-distribution fairness learning methods suffer significant performance drop under unseen distribution shifts. Specifically, we train the model using normal training, AdvDebias (Zhang et al., 2018) and LAFTR (Madras et al., 2018) on Adult (Kohavi, 1996) dataset (i.e., US Census data before 1996). We evaluate the $\Delta_{EO}$ on the original Adult test set and the 2015 subset of Folktables datase (Ding et al., 2021) (i.e., US Census data in 2015) respectively, in order to check whether the fairness achieved on in-distribution data is preserved under the temporal distribution shift. The results are shown in Figure 3.

As we can see, LAFTR and AdvDebias successfully improve the in-distribution fairness compared with normal training. However, both methods suffer significant performance drop in terms of $\Delta_{EO}$ and $\Delta_{EOpp}$ under the temporal distribution shift. Moreover, under the distribution shift, the $\Delta_{EO}$ and $\Delta_{EOpp}$ achieved by LAFTR and AdvDebias are almost the same with or even worse than that of normal training. In other words, the models trained by LAFTR and AdvDebias are almost as unfair as a normally trained model under this naturally occurring distribution shift.

## 4    Curvature Matching: Towards Robust Fairness under Unseen Distributional Shifts

In this section, we present our proposed solution for the robust fairness challenge described in Section 3.

### 4.1    Loss Curvature as the Measure for Robustness

Before introducing our robust fairness learning method, we need to first define the measure for model robustness under unseen distributional shifts.

Consider a binary classifier $f(\cdot; \theta)$ trained on two groups of data $X_1$ and $X_2$. Our goal is to define a metric to measure the gap of model robustness between the two groups. Previous research (Guo et al., 2018; Weng et al., 2018) has shown both theoretically and empirically that deep model robustness scales with its model smoothness.[1] Motivated by the above, we use the spectral norm of Hessian matrix to approximate local smoothness as a measure of model robustness. Specifically, given an input $x$, the Hessian matrix $H(x)$ is defined as the second-order gradient of $\mathcal{L}(x)$ with respect to model weights $\theta$: $H(x) = \nabla_\theta^2 \mathcal{L}(x)$. The approximated local curvature $\mathcal{C}(x)$ at point $x$ is thus defined as:

$$\mathcal{C}(x) = \sigma(H(x)), \tag{3}$$

where $\sigma(H)$ is the spectral norm (SN) of $H$: $\sigma(H) = \sup_{v:\|v\|_2=1} \|Hv\|_2$. Intuitively, $\mathcal{C}(x)$ measures the maximal directional curvature or change rate of the loss function at $x$. Thus, smaller $\mathcal{C}(x)$ indicates better local smoothness around $x$.

**Practical Curvature Approximation**   It is inefficient to directly optimize the loss curvature through Eq. (3), since it involves high order gradients.[2] To solve this problem, we use a one-shot power iteration method (PIM) for practical approximation of $\mathcal{C}(x)$ during training. First we rewrite $\mathcal{C}(x)$ with the following form: $\mathcal{C}(x) = \sigma(H(x)) = \|H(x)v\|$, where $v$ is the dominant eigenvector with the maximal eigenvalue, which can be calculated by power iteration method. In practice, we estimate the dominant eigenvector $v$ by the gradient direction: $\tilde{v} := \frac{\text{sign}(g)}{\|\text{sign}(g)\|} \approx v$, where $g = \nabla_\theta \mathcal{L}(x)$. This is because previous works have observed a large similarity between the dominant eigenvector and the gradient direction (Miyato et al., 2018; Moosavi-Dezfooli et al., 2019). We further approximate Hessian matrix by finite differentiation on gradients:

---

[1]We mainly focus on covariate shift since the curvature smoothness is related to robustness under covariate shift. Our experiments on CelebA and C&C datasets verified the effectiveness of our method under covariate shift. For example, on CelebA dataset, the source (training) distribution contains young faces and the target (test) distribution contains old faces, which is a covariate shift setting. Our method is also empirically effective against temporal shift, as verified in our experiments on Adult dataset.

[2]The Hessian matrix itself involves second order gradients, and backpropagation through Eq. (3) requires even higher order gradient on top of the Hessian matrix SN.

$H(x)v \approx \frac{\nabla_\theta \mathcal{L}(x+hv) - \nabla_\theta \mathcal{L}(x)}{h}$ where $h$ is a small constant. As a result, the final approximation of curvature smoothness is

$$\mathcal{C}(x) \approx \frac{\|\nabla_\theta \mathcal{L}(x + h\tilde{v}) - \nabla_\theta \mathcal{L}(x)\|}{|h|}. \tag{4}$$

## 4.2 Curvature Matching

Equipped with the practical curvature approximation, now we can match the curvature distribution of the two groups by minimizing their maximum-mean-discrepancy (MMD) (Gretton et al., 2012) distance. Suppose $\mathcal{C}(X_1) \sim \mathcal{Q}_1$ and $\mathcal{C}(X_2) \sim \mathcal{Q}_2$, where $\tilde{\mathcal{C}}(\cdot)$ is defined in Eq. 4, we define the curvature matching loss functions as:

$$\mathcal{L}_{cm} = \text{MMD}^2(\mathcal{Q}_1, \mathcal{Q}_2), \tag{5}$$

The MMD distance, which is widely used to measure the distance between two high-dimensional distributions in deep learning (Li et al., 2015; 2017; Bińkowski et al., 2018), is defined as

$$\text{MMD}^2(\mathcal{P}, \mathcal{Q}) = \mathbb{E}_\mathcal{P}[k(X, X)] - \\ 2\mathbb{E}_{\mathcal{P},\mathcal{Q}}[k(X, Y)] + \mathbb{E}_\mathcal{Q}[k(Y, Y)] \tag{6}$$

where $X \sim \mathcal{P}$, $Y \sim \mathcal{Q}$ and $k(\cdot, \cdot)$ is the kernel function. In practice, we use finite samples from $\mathcal{P}$ and $\mathcal{Q}$ to statistically estimate their MMD distance:

$$\text{MMD}^2(\mathcal{P}, \mathcal{Q}) = \frac{1}{M^2} \sum_{i=1}^{M} \sum_{i'=1}^{M} k(x_i, x_{i'})$$
$$- \frac{2}{MN} \sum_{i=1}^{M} \sum_{j=1}^{N} k(x_i, y_j) + \frac{1}{N^2} \sum_{j=1}^{N} \sum_{j'=1}^{N} k(y_j, y_{j'}) \tag{7}$$

where $\{x_i \sim \mathcal{P}\}_{i=1}^{M}$, $\{y_j \sim \mathcal{Q}\}_{j=1}^{N}$, and we use the mixed RBF kernel function as $k(x, y)$. Selecting a proper kernel width is important for RBF kernel (Sutherland et al., 2016; Liu et al., 2020). We follow Li et al. (2015; 2017) to use a series of different kernel width simultaneously: $k(x, y) = \sum_{\sigma \in \mathbb{S}} e^{-\frac{\|x-y\|^2}{2\sigma^2}}$ with $\mathbb{S} = \{1, 2, 4, 8, 16\}$. During training, we use the whole batch of training samples to estimate MMD, where the batch size is 64. In other words, $M + N = 64$ in equation 7.

As a side note, MMD has been previously used in fairness learning. Quadrianto & Sharmanska (2017) defines a more general fairness metric using MMD distance, and shows DP and EO to be the special cases of their unified metric. Their paper, however, still focuses on the in-distribution fairness. In contrast, our CUMA minimizes the MMD distance on the curvature distributions to achieve robust fairness.

Back to our method. After defining $\mathcal{L}_{cm}$, we add it to the traditional adversarially fair training (Ganin et al., 2016; Madras et al., 2018) loss function as a regularizer, in order to attain both in-distribution fairness and robust fairness. As illustrated in Figure 2, our model follows the same "two-head" structure as traditional adversarial learning frameworks (Ganin et al., 2016; Madras et al., 2018), where $h_t$ is the utility head for the target task, $h_a$ is the adversarial head to predict sensitive attributes, and $f_s$ is the shared backbone.[3] Suppose for each sample $x_i$, the sensitive attribute is $a_i$ and the corresponding target label is $y_i$, then our overall optimization problem can be written as:

$$\min_{\theta_s, \theta_t} \max_{\theta_a} \mathcal{L} = \min_{\theta_s, \theta_t} \max_{\theta_a} (\mathcal{L}_{clf} - \alpha\mathcal{L}_{adv} + \gamma\mathcal{L}_{cm}) \tag{8}$$

---

[3]Thus the binary classifier $f(\cdot; \theta) = h_t(f_s(\cdot; \theta_s); \theta_t)$, with $\theta = \theta_t \cup \theta_s$.

where

$$\mathcal{L}_{clf} = \frac{1}{N} \sum_{i=1}^{N} \ell(h_t(f_s(x_i; \theta_s); \theta_t), y_i), \tag{9}$$

$$\mathcal{L}_{adv} = \frac{1}{N} \sum_{i=1}^{N} \ell(h_a(f_s(x_i; \theta_s); \theta_a), a_i), \tag{10}$$

$\ell(\cdot, \cdot)$ is the cross-entropy loss function, $\alpha$ and $\gamma$ are trade-off hyperparameters, and $N$ is the number of training samples.

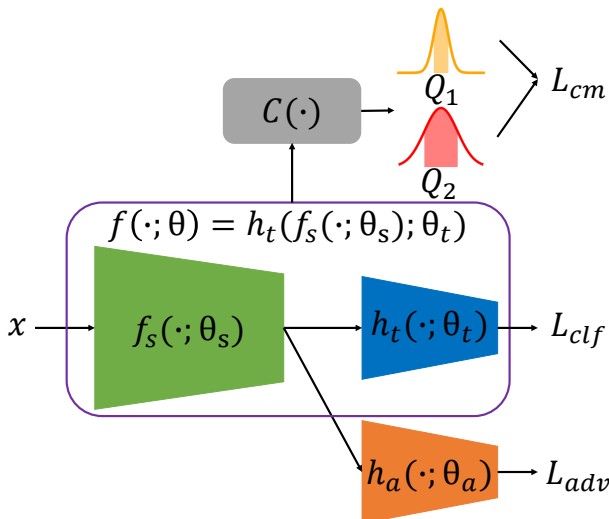

Figure 2: The overall framework of CUMA. $x$ is the input sample. $h_t$ is the utility head for the target task. $h_a$ is the adversarial head to predict sensitive attributes. $f_s$ is the shared backbone. $\mathcal{C}(\cdot)$ is the curvature estimation function, as defined in Eq. (3). $\mathcal{Q}_1$ and $\mathcal{Q}_2$ are local curvature distributions of majority and minority groups, respectively. $\mathcal{L}_{cm}$, $\mathcal{L}_{clf}$ and $\mathcal{L}_{adv}$ are three loss terms as defined in Eq. (5), (9), and (10), respectively.

## 5 Experiments

### 5.1 Experimental Setup

**Datasets and pre-processing**  Experiments are conducted on three datasets widely used to evaluate machine learning fairness: Adult (Kohavi, 1996), and CelebA (Liu et al., 2015), and Communities and Crime (C&C) (Redmond & Baveja, 2002).[4] Adult dataset has 48,842 samples with basic personal information such as education and occupation, where 30,000 are used for training and the rest for evaluation. The target task is to predict the person's annual income, and we use "gender" (male or female) as the sensitive attribute. The features in Adult dataset are of either continuous (e.g., age) or categorical (e.g. sex) values. We use one-hot encoding on the categorical features and then concatenate them with the continuous ones. We use data whitening on the concatenated features. CelebA has over 200,000 images of celebrity faces, with 40 attribute annotations. The target task is to predict gender (male or female) and the sensitive attribute to protect is "chubby". We randomly select $10,000$ as training samples and $1,000$ as testing samples. All images are center-cropped and resized to $64 \times 64$, and pixel values are scaled to $[0, 1]$. C&C dataset has 1,994 samples with neighborhood population statistics, where 1,500 are used for training and the rest for evaluation. The target task is to predict violent crime per capita, and we use "RacePctBlack" (percentage of black population in the neighborhood) as sensitive attributes. All features in C&C dataset are of continous values in $[0, 1]$. To

---

[4]Traditional image classification datasets (e.g., ImageNet) are not directly applicable since they lack fairness attribute labels.

fit in the fairness problem setting, we binarilize the target and sensitive attributes with the top-30% largest value as the threshold (As a result $P[A = 0] = 30\%$ and $P[Y = 0] = 30\%$). We also do data-whitening on C&C.

**Models**  For C&C and Adult datasets, we use two-layer MLPs for $f_s$, $h_t$ and $h_a$. Specifically, suppose the input feature dimension is $d$, then the dimensions of hidden layers in $f_s$ and $h_t$ are $d \to 100 \to 64$ and $64 \to 32 \to 2$, respectively. $h_a$ has identical model structure with $h_t$. For all three sub-networks, ReLU activation function and dropout layer with 0.25 dropout ratio are applied between the two fully connected layers. For CelebA dataset, we use ResNet18 as backbone, where the first three stages are used as $f_s$ and the last stage (together with the fully connected classification layer) is used as $h_t$. The auxiliary adversarial head $h_a$ has the same structure as $h_t$.

**Baseline methods**  We compare CUMA with the following state-of-the-art in-distribution fairness algorithms. Adversarial debiasing (AdvDebias) (Zhang et al., 2018) is one of the most popular fair training algorithm based on adversarial training (Ganin et al., 2016). Madras et al. (2018) proposes a similar framework termed Learned Adversarially Fair and Transferable Representations (LAFTR), by replacing the cross-entropy loss used in (Zhang et al., 2018) with a group-normalized $\ell_1$ loss, which is shown to work better on highly unbalanced datasets. We also include normal (fairness-ignorant) training as a baseline.  We also add the "Oracle" baseline method: We train with AdvDebias on the target distribution and tested the model on the target distribution, to see how hard it is to directly achieve fairness on the target data distribution. Lastly, we also add RobustFair(Mandal et al., 2020), a previous method to pursue fairness beyond the training data, as a baseline.

**Evaluation metric**  We report the overall accuracy on all test samples in the original test sets. To measure in-distribution fairness, we use $\Delta_{EO}$ and $\Delta_{EOpp}$ on the original test sets. To measure robust fairness under distribution shifts, we use $\Delta_{EO}$ and $\Delta_{EOpp}$ on test sets with distribution shifts. See the following paragraph for the details in constructing distribution shifts. Since there is inherent trade-off between fairness and accuracy, we report the trade-offs between fairness and accuracy by setting the loss function weights (e.g., $\alpha$ and $\gamma$) to different values. For example, the larger $\alpha$, the better fairness and the worse accuracy. We show the results by visualizing the trade-off curves between fairness and accuracy of different methods. The closer the curve to the top-left corner (i.e., with larger accuracy and smaller $\Delta_{EO}$ or $\Delta_{EOpp}$ ), the better Pareto frontier is achieved.

**Distribution shifts**  Adult dataset contains US Census data collected before 1996. We use the 2014 and 2015 subset of Folktables dataset (Ding et al., 2021), which contain US Census data collected in 2014 and 2015 respectively, as the test sets with distribution shifts. This simulates the real-world temporal distributional shifts.

On CelebA dataset, we train the model on $10,000$ "young" face images. We use another $1,000$ "young" face images as in-distribution test set and $1,000$ "not young" face images as the test set with distribution shifts. This simulates the real-world scenario when the model is trained and used on people with different age groups. We also construct another test set with a different type of distribution shift, by applying strong JPEG compression on the original $1,000$ "young" test images, following Hendrycks & Dietterich (2019). This simulates the scenario when the model is trained on good quality images while the test images has poor visual quality.

For C&C dataset, we construct two artificial distribution shifts by adding random Gaussian and uniform noises, respectively, to the test data. Specifically, the categorical features in C&C dataset are first one-hot encoded and then whitened into float-value vectors, where noises are added. Both types of noises have mean $\mu = 0$ and has standard derivation $\sigma = 0.03$ .

**Implementation details**  Unless further specified, we set the loss trade-off parameter $\alpha$ to 1 in all experiments by default. We use Adam optimizer (Kingma & Ba, 2014) with initial learning rate $10^{-3}$ and weight decay $10^{-5}$. The learning rate is gradually decreased to 0 by cosine annealing learning rate scheduler (Loshchilov & Hutter, 2016). On both Adult and C&C datasets, we train for 50 epochs from scratch for all methods. On CelebA dataser, we first normally train a model for 100 epochs, and then finetune it for 20

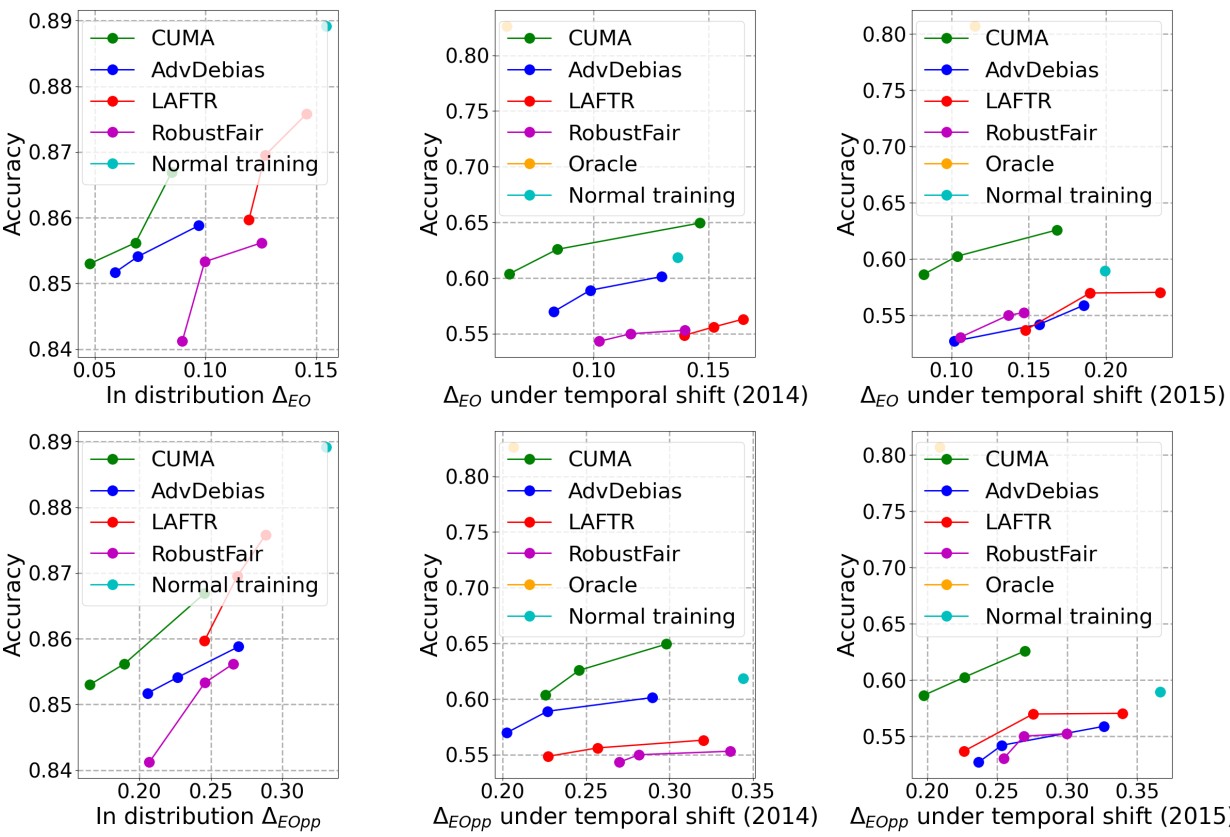

Figure 3: Trade-off curves between fairness and accuracy of different methods. Results are reported on Adult dataset with "Sex" as the sensitive attribute.

epochs using CUMA. For fair comparison, we train for 120 epochs on CelebA for all baseline methods. The constant $h$ in Eq. (4) is set to 1 by default.

## 5.2 Main Results

Experimental results on three datasets with different sensitive attributes are shown in Figures 3, 4, 5, where we compare CUMA with the baseline methods on different metrics as discussed in Section 5.1. "Normal" means standard training without any fairness regularization. All numbers are shown as percentages. Many intriguing findings can be concluded from the results.

First, we see that previous state-of-the-art fairness learning algorithms would be jeopardized if distributional shifts are present in test data. For example, on Adult dataset (Figure 3), LAFTR achieves $\Delta_{EO} = 11.96\%$ on in-distribution test set, while that number is increased to 13.95% on the 2014 test set and 14.80% on the 2015 test set, which is almost as unfair as the normally trained model. Similarly, on CelenA dataset with "Chubby" as the sensitive attribute (Figure 4), LAFTR achieves $\Delta_{EO} = 31.02\%$ on the original CelebA test set, while that number is increased to 35.88% and 37.46% under distribution shifts of user age and image quality, respectively.

Second, we see that CUMA achieves the best robust fairness under distribution shifts under all evaluated settings, while maintaining similar in-distribution fairness and overall accuracy. For example, on Adult dataset (Figure 3), CUMA achieves 1.92% and 1.96% less $\Delta_{EO}$ than the second-best performer (AdvDebias) on the 2014 and 2015 Census dataset, respectively. On CelebA dataset (Figure 4) with "Chubby" as the sensitive attribute, CUMA achieves 2.62% and 2.74% less $\Delta_{EO}$ than the second-best performers under distribution shifts of user age and image quality, respectively. Moreover, still in Figure 4, CUMA and LAFTR achieve almost identical in-distribution fairness (the difference between their $\Delta_{EO}$ on original test set is 0.5%), CUMA keeps the fairness under distribution shifts (with only around 1.6% increase in $\Delta_{EO}$), while the fairness

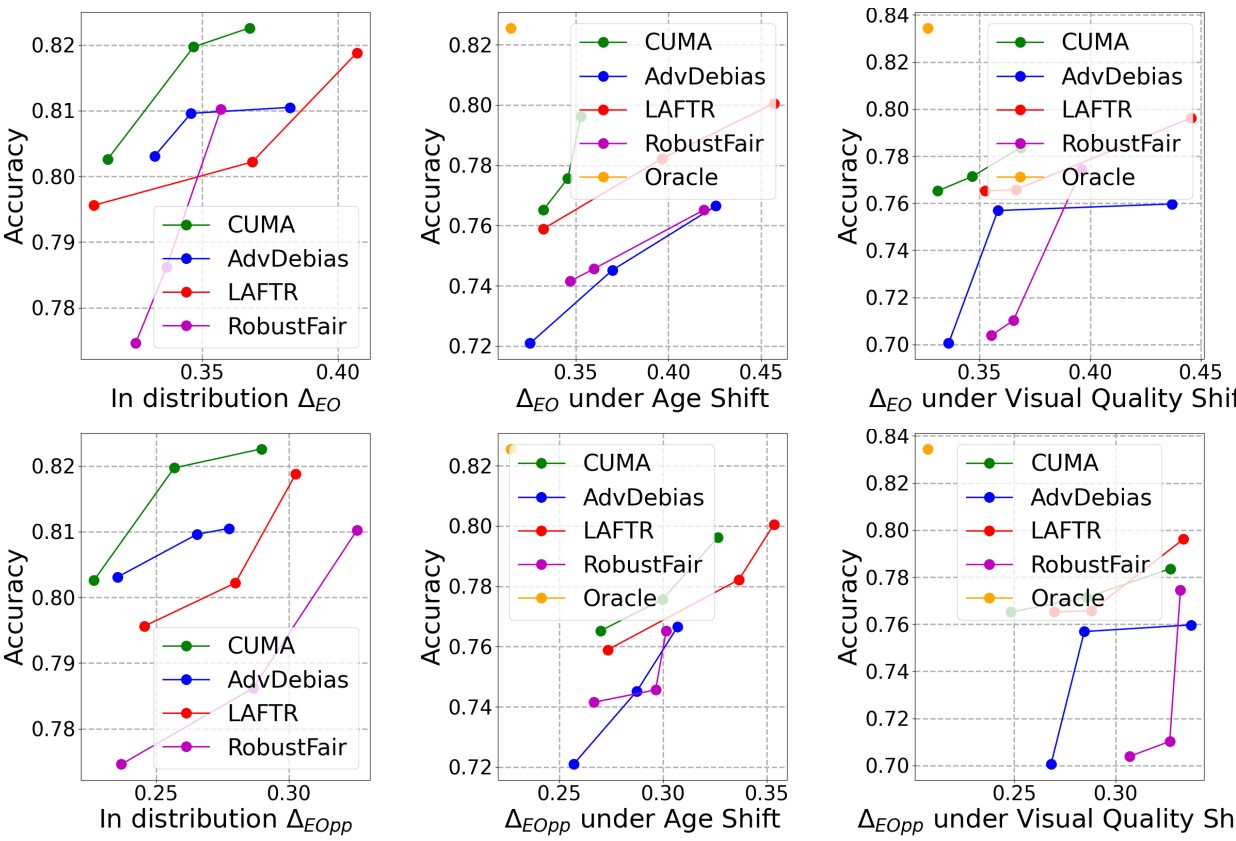

Figure 4: Trade-off curves between fairness and accuracy of different methods. Results are reported on CelebA dataset with "Chubby" as the sensitive attribute.

achieved by LAFTR is significantly worse, especially under the image quality distribution shift, where the $\Delta_{EO}$ is increased by 6.44%.

### 5.3 Ablation Study

In this section, we check the sensitivity of CUMA with respect to its hyper-parameters: the loss trade-off parameters $\alpha$ and $\gamma$ in Eq. (8) and $h$ in Eq. (4). Results are shown in Table 1. When fixing $\alpha = 1$, the best trade-off between overall accuracy and robust fairness is achieved at round $\gamma = 1$, which we use as the default $\gamma$. Varying the value of $h$ hardly affects the performance of CUMA.

Table 1: Ablation study results on the loss trade-off parameters $\alpha$ and $\gamma$ in the CUMA algorithm. Results are reported on C&C dataset with "RacePctBlack" as the sensitive attribute.

|  | $\alpha$ | | | | $\gamma$ | | | | $h$ | |
|---|---|---|---|---|---|---|---|---|---|---|
|  | 0 | 0.1 | 1 | 10 | 0 | 0.1 | 1 | 10 | 0.1 | 1 |
| Accuracy | 89.21 | 86.94 | 85.40 | 83.75 | 84.79 | 85.19 | 85.40 | 84.79 | 85.32 | 85.40 |
| $\Delta_{EO}$ with Gaussian noise | 64.27 | 66.51 | 28.74 | 33.16 | 39.84 | 38.85 | 28.74 | 27.95 | 30.56 | 28.74 |

## 6 Conclusion

In this paper, we first observe the challenge of robust fairness: Existing state-of-the-art in-distribution fairness learning methods suffer significant performance drop under unseen distribution shifts. To solve this problem, we propose a novel robust fairness learning algorithm, termed Curvature Matching (CUMA), to simultaneously

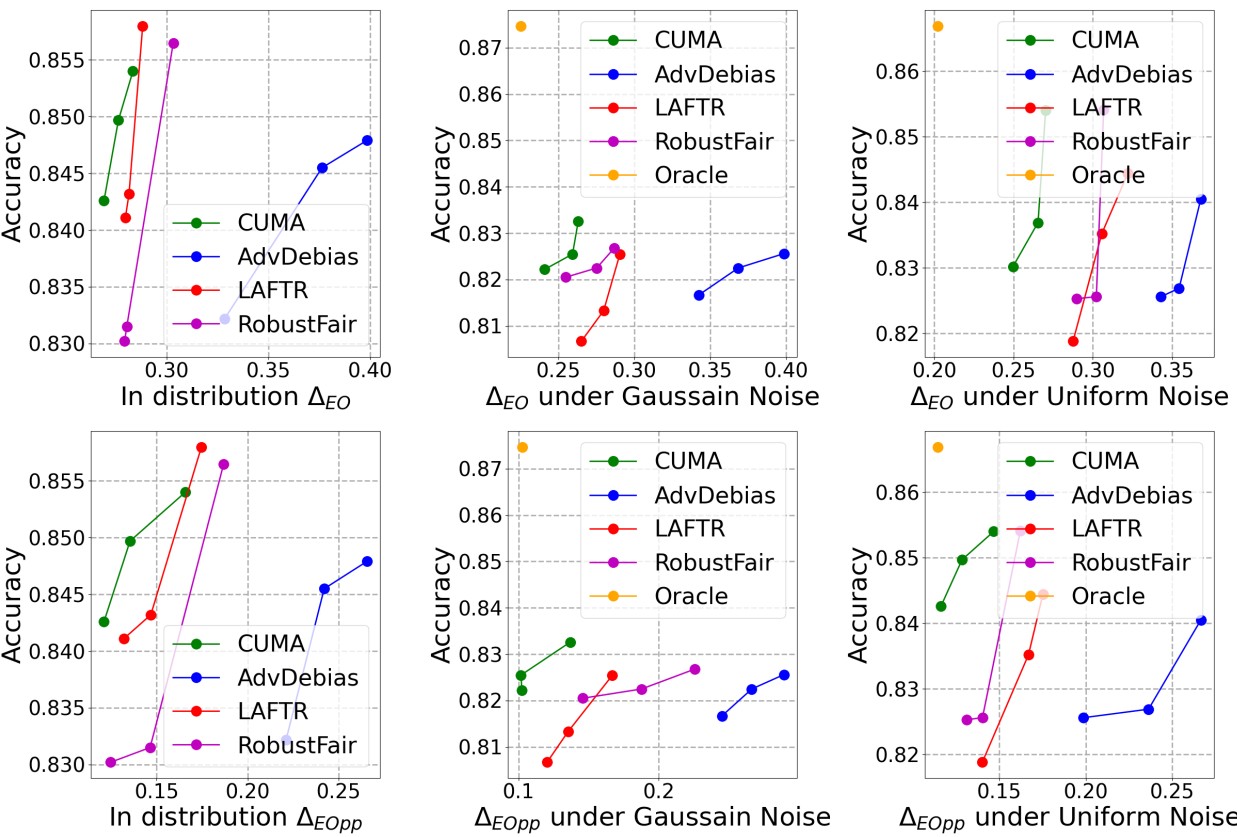

Figure 5: Trade-off curves between fairness and accuracy of different methods. Results are reported on C&C dataset with "RacePctBlack" as the sensitive attribute.

achieve both traditional in-distribution fairness and robust fairness. Experiments show CUMA achieves more robust fairness under unseen distribution shifts, without more sacrifice on either overall accuracies or the in-distribution fairness compared with traditional in-distribution fairness learning methods.

# 7 Broader Impact

With the wide application of machine learning models in many real-world applications, new requirements have been raised to guarantee machine learning models make ethical and fair decisions. This work takes one step forward in this direction, by generalizing the fairness on in-distribution data to out-of-distribution data.

**Limitations and Future Works** Following previous fairness learning works (Zhang et al., 2018; Madras et al., 2018), this paper focuses on the binary classification problem with binary privacy attributes. The proposed method can potentially be generalized to the multiclass classification or continuous prediction problems following similar ideas as in (Mary et al., 2019; Denis et al., 2021), which we leave as the future work. We also empirically find that the proposed method do not directly benefit model accuracy under distributional shifts. It would be a good future direction to investigate how to simultaneously achieve better model accuracy and fairness under distributional shifts. Another topic we leave as future work is to study the theoretical guarantees on the generalization bounds for fairness.

**Acknowledgement** The work of J. Zhou is in part supported by the National Science Foundation under Grant IIS-2212174, IIS-1749940, Office of Naval Research N00014-20-1-2382, and National Institute on Aging (NIA) RF1AG072449. The work of Z. Wang is in part supported by the National Science Foundation under Grant IIS-2212176.

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
