# OpenReview forum: "How Robust is Your Fairness? Evaluating and Sustaining Fairness under Unseen Distribution Shifts"
_TMLR — Accepted by TMLR_

### Review · Reviewer_F7Tn · 2023-01-04

**Summary Of Contributions:**

This paper studies the robustness problem of fairness under unknown distribution shifts. It shows some adversarial-training-based fairness learning methods are non-robust to distribution shifts.  To address the problem, by introducing the loss curvature as the measure for robustness,  the paper proposes a curvature matching method to make the robustness of the two data groups similar. Experiments verify the effectiveness of the proposed method compared with two fairness learning methods.

**Audience:**

Yes

**Claims And Evidence:**

Yes

**Requested Changes:**

See the above weaknesses.

**Strengths And Weaknesses:**

**Strengths**
- The research problem is interesting and realistic. The example of the unknown distribution shifts in self-driving is clear.
- The objective that the minority and majority groups should have similar robustness is reasonable. The technique is sound and easy to follow.
- The design of the experiments for producing distribution shifts is reasonable and practical.

**Weaknesses**
- The paper claims that the fairness achieved by existing methods can be easily broken by slight distribution shifts, while only two adversarial-training-based fairness learning methods (LAFTR and AdvDebias) are used to verify and compared.  I think more different and recent fairness learning methods should be compared to support this claim. Besides,  since multiple fairness criteria have been defined as denoted in the paper, it would be better to also report some other fairness criteria in the experiments except  Equalized Odds to support the claim.
- It would be better to give some experimental evidence about the effectiveness of practical curvature approximation since this effectiveness might influence the learning objective a lot.
- Can the curvature matching perform as a plug-and-play module added to other recent fairness learning methods?  If so, I think this will arise a lot of interest for some readers.

---

> ### Author Response · Authors · 2023-02-03
> **Response to Reviewer F7Tn**
>
> Thank you for your insightful suggestions and careful reading. We have carefully addressed your concerns and suggestions below.
>
> 1. Add more baseline methods and other fairness metrics.
>
> A: Thanks for your suggestion. We have added [3] as a new baseline method and Equalized Opportunity as a new metric. Please check the updated Figure 3 for details.
>
> [3] Ensuring Fairness Beyond the Training Data. NeurIPS, 2020.
>
>
>
> 2. What is the effectiveness of practical curvature approximation?
>
> A: As mentioned in Section 4.1, our curvature approximation method is similar with that used in (Moosavi-Dezfooli et al., 2019), in which the effectiveness of this approximation is verified.
>
> 3. Can the curvature matching perform as a plug-and-play module added to other recent fairness learning methods?
>
> A: Thanks for the insightful comments. By design, our method should be able to be added as a plug-and-play module to other in-processing fairness learning methods. We leave the empirical verification of this point as our future work.

---

> ### Comment · Action_Editors · 2023-02-16
> **Recommendation**
>
> Dear Reviewer F7Tn,
>
> Could you take a look at the rebuttal and submit your official recommendation by the end of Feb 17 (Friday)?
>
> AE

---

### Review · Reviewer_QVkJ · 2023-01-04

**Summary Of Contributions:**

In this paper, the authors considered the robust fairness problem under the unseen distribution shift and proposed a method for this problem by matching the loss curvature distributions of the majority and minority groups to let the model have the same generalization ability for the two groups. The method is evaluated on three datasets under the fairness settings. The main contributions of the paper include the proposed fairness robustness method and the related empirical evaluations.

**Audience:**

Yes

**Broader Impact Concerns:**

The paper does not include a section for describing the broader impact but personally I do not have big concerns regarding the broader impact for this work.


**Claims And Evidence:**

Yes

**Requested Changes:**

1. I may suggest the authors explain more about their assumptions on the distribution shift (weakness point 2).
2. It is better if the proposed method can be compared with baselines that consider distribution shift if possible (weakness point 3).
3. The authors should carefully check the format of the references to keep them consistent (weakness point 4).
4. In the ablation study, it is better to check for example the cases when 1) both $\alpha$ and $\gamma$ are zeros 2) only $\alpha=0$ 3) only $\gamma=0$ , to see the contributions of the different loss terms to the final results.

**Strengths And Weaknesses:**

Strengths:
1. The research problem, robust fairness issues under the distribution shift, is quite practical and interesting to study.
2. The proposed method does not need the target labeled/unlabeled data to have the robustness, which is claimed as one of the important differences from prior works.
3. The writings of the paper is quite easy to follow and understand.

Weaknesses:
1. The novelty of the proposed method may not be very large. The proposed method is largely based on prior ideas of curvature matching and adversarial training.
2. I'm wondering which kind of distribution shift could the proposed method handle? The authors only slightly mentioned that they focus on covariate shift in Sec 2.1. If it is true, why does the covariate shift assumption necessary in this work? How to ensure this assumption in the experiments? I think at least the authors should explain more clearly about their assumption on distribution shift in the paper.
3. The proposed method is compared with the baselines that only considers the in-distribution fairness. It makes more sense to compare it with some robust fairness methods under distribution shift, for example, Mandal et al. Ensuring fairness beyond the training data. In NeurIPS 2020.
4. Format of the references is not consistent. Take NeurIPS papers in the references as an example:
- Debmalya Mandal, Samuel Deng, Suman Jana, and Daniel Hsu. Ensuring fairness beyond the training data. In NeurIPS, 2020.
- Yiwen Guo, Chao Zhang, Changshui Zhang, and Yurong Chen. Sparse DNNs with improved adversarial robustness. In Advances in Neural Information Processing Systems, 2018.
- Flavio P Calmon, Dennis Wei, Bhanukiran Vinzamuri, Karthikeyan Natesan Ramamurthy, and Kush R Varshney. Optimized pre-processing for discrimination prevention. In International Conference on Neural Information Processing Systems, pp. 3995–4004, 2017.

---

> ### Author Response · Authors · 2023-02-03
> **Response to Reviewer QVkJ**
>
> Thank you for your insightful suggestions and careful reading. We have carefully addressed your concerns and suggestions below.
>
> 1. Explain more clearly about their assumption on distribution shift.
>
> A: We mainly focus on covariate shift since the curvature smoothness is related to robustness under covariate shift. Our experiments on CelebA and C&C datasets verified the effectiveness of our method under covariate shift. For example, on CelebA dataset, the source (training) distribution contains young faces and the target (test) distribution contains old faces, which is a covariate shift setting. Our method is also empirically effective against temporal shift, as verified in our experiments on Adult dataset. We have added this discussion in Section 4.1.
>
> 2. Compare with “Ensuring fairness beyond the training data”.
>
> A: Thanks for pointing out this related work. We have added it as a baseline in the updated Figure 3.
>
> 3. Format of the references is not consistent.
>
> A: Thank you for your careful reading. We have aligned the reference formats.
>
> 4. Ablation study for $\alpha=0$ and $\gamma=0$.
>
> A: Thank you for your suggestion. The results are added in Table 4.

---

### Review · Reviewer_cMfQ · 2023-01-16

**Summary Of Contributions:**

This paper is novel from problem setting and technical solutions.

1. In the view of significance to the community/society, this paper considers an interesting problem: maintaining fairness in unseen domains. To the best of my knowledge, this is new and novel. More importantly, this problem is worthy to investigate because that distribution shift happens all the time. It is glad to see a study regarding the distribution shift in the fairness field.

2. In the view of technical contributions, this paper studies this novel problem from Loss Curvature, which was found to be useful in robustness by Guo et al. (2018) and Weng et al. (2018). The motivation is clear and interesting. However, we cannot directly use this idea. To this end, this paper considers Curvature Matching which is new to the field.

**Audience:**

Yes

**Broader Impact Concerns:**

There are no concerns regarding broader impact.

**Claims And Evidence:**

Yes

**Requested Changes:**

Cons:

1. Experiment setting should be clarified. I am not sure where only two groups are considered in this paper. If yes, more motivations or explanations should be given.

2. Bandwidth selection of MMD is very important [r1, r2], more discussion should be added.

3. Eq. (3) is not clear. Parameters should be linked to each loss function here.

[r1] Generative models and model criticism via optimized maximum mean discrepancy, ICLR.

[r2] Learning Deep Kernels for Non-Parametric Two-Sample Tests, ICML.

**Strengths And Weaknesses:**

Pros:

1. In the view of significance to the community/society, this paper considers an interesting problem: maintaining fairness in unseen domains. To the best of my knowledge, this is new and novel. More importantly, this problem is worthy to investigate because that distribution shift happens all the time. It is glad to see a study regarding the distribution shift in the fairness field.

2. In the view of technical contributions, this paper studies this novel problem from Loss Curvature, which was found to be useful in robustness by Guo et al. (2018) and Weng et al. (2018). The motivation is clear and interesting. However, we cannot directly use this idea. To this end, this paper considers Curvature Matching which is new to the field.

3. This paper is easy to follow and understand.

Cons:

1. Experiment setting should be clarified. I am not sure where only two groups are considered in this paper. If yes, more motivations or explanations should be given.

2. Bandwidth selection of MMD is very important [r1, r2], more discussion should be added.

3. Eq. (3) is not clear. Parameters should be linked to each loss function here.

[r1] Generative models and model criticism via optimized maximum mean discrepancy, ICLR.

[r2] Learning Deep Kernels for Non-Parametric Two-Sample Tests, ICML.

---

> ### Author Response · Authors · 2023-02-03
> **Response to Reviewer cMfQ**
>
> Thank you for your insightful suggestions and careful reading. We have carefully addressed your concerns and suggestions below.
>
> 1. Experiment setting should be clarified.
>
> A: Yes, you are correct that we focus on the binary classification problem with binary privacy attributes in this paper. Such problem setting is popular in machine learning fairness, since most fairness learning metrics like equalized odds and equalized opportunity are defined under such problem setting. Specifically, our problem setting follows previous fairness learning works (Zhang et al., 2018; Madras et al., 2018). The proposed method can potentially be generalized to the multiclass classification or continuous prediction problems following similar methods as in (Mary et al., 2019; Denis et al., 2021), which we leave as the future work. We added discussion on this limitation in Section 7 using blue texts.
>
> 2. More discussion on MMD bandwidth.
>
> A: Thanks for pointing out the related works. We have added them in Section 2 using blue texts. In our method, we used a series of kernel widths: S = {1, 2, 4, 8, 16}, following previous works [4,5], as described in Section 4.3.
>
> [4] Generative moment matching networks.
>
> [5] MMD GAN: Towards deeper understanding of moment matching network.
>
> 3. Eq. (3) is not clear. Parameters should be linked to each loss function here.
>
> A: Thanks for your suggestion. Eq. (3) is used to calculate the loss term in Eq. (4). We have added this description when defining Eq. (4) at the beginning of Section 4.2 using blue texts.

---

> ### Comment · Action_Editors · 2023-02-16
> **Recommendation**
>
> Dear Reviewer cMfQ,
>
> Could you take a look at the rebuttal and submit your official recommendation by the end of Feb 17 (Friday)?
>
> AE

---

### Review · Reviewer_4fit · 2023-01-21

**Summary Of Contributions:**

This paper studies a very important problem: How well does fairness mitigation generalize out-of-distribution. The authors build on the heuristic that a smoother loss landscape generalizes better out-of-distribution and add a regularizer to improve the generalization of the adversarial training method for fairness. The authors also curate three datasets to extend existing popular fairness datasets out of distribution. It is shown that for equalized odds notion of fairness, the proposed method results in a better tradeoff curve between fairness violation and accuracy on these binary classification tasks with binary sensitive attributes.

**Audience:**

Yes

**Broader Impact Concerns:**

The paper is missing a broader impact section that clearly discusses the limitations.

**Claims And Evidence:**

No

**Requested Changes:**

Overall, while the paper is studying a very important problem, I think the contributions of the paper as it stands falls short of a fully fledged paper. I am proposing some axes for improving the paper for it to become a valuable contribution to the literature.

- Given the datasets are an important contribution of this paper, they warrant more scrutiny and description. For example, what if you trained on the new datasets and tested on it as an Oracle method to determine how hard these benchmarks are? Also, would it have been possible to develop other benchmarks based on the well-known distribution shift datasets like WILDS, Imagenet-C, etc?

- SOTA in-processing fairness methods are capable of producing fairness violation / accuracy tradeoff curves rather than a single point. Please turn all results in comparing tradeoff curves, which will be much more informative, and in particular the goal should be to achieve a tradeoff curve that dominates the baselines. What has been reported in Table 1 misses the point completely because it doesn't even consider the performance. Please turn all these comparisons to tradeoff curves.

- Please discuss how the kernel width has been selected for the MMD loss. Also, please discuss how many samples are needed for the MMD loss to be representative of the distribution mismatch between the curvatures.

- Please expand the scope of the effectiveness of the proposed method along the *methods* and *fairness notion* and *scaling* axes. As it stands the experiments only consider adversarial training for equalized odds in binary classification with binary sensitive attributes which makes the applicability of the scope of this paper too narrow.

- Please add a broader impact section that clearly discusses the limitations of the methods used in this paper, given the methods might be used in socially consequential applications.

minor issues and typos:

- The specific subsection on fairness in CV seems totally out of place because I didn't find anything in the rest of the paper to warrant that distinction. Please blend it in with the rest of the subsections. Also, please refer to [Szabó et al., 2021] on fairness in CV.

Szabó, Attila, Hadi Jamali-Rad, and Siva-Datta Mannava. "Tilted cross-entropy (TCE): Promoting fairness in semantic segmentation." In Proceedings of the IEEE/CVF Conference on Computer Vision and Pattern Recognition, pp. 2305-2310. 2021.

- page 1, 2nd paragraph
> ... commonly encounter data ...

, -> "... commonly encounters data ..."

- page 3, Section 2
> Compared with individual fairness, group fairness is a more popular setting and thus the focus of our paper"

The notion of fairness that is used is determined by the application at hand; some applications require individual fairness while others may warrant group fairness. It is okay to focus on group fairness in this paper, but I would avoid calling it a more popular notion of fairness.

- page 4, Section 3
> and LAFTR Madras et al. (2018) on Adult Kohavi (1996) dataset

Please make sure to use \cite, \citet, \citep properly. This is a repeated issue throughout the paper.

- page 6, Section 4.2
>  DP and EO to be the spatial cases of their

special

**Strengths And Weaknesses:**

Strengths

- The problem that is studied in this paper is extremely important to the community.

- The paper constructs distribution shift datasets on three popular fairness datasets. These datasets could be of independent interest to the community.

- The proposed method is simple and based on intuitive principle that the distribution of the curvatures of losses should be similar on both subgroups. The experiments also show effectiveness of the method.

Weaknesses

- The cited literature in the paper is outdated and only covers papers published through 2020. I encourage the authors to update the literature survey and cover the papers published in the past two years as well. You may find this survey helpful to this end:

Hort, Max, et al. "Bias mitigation for machine learning classifiers: A comprehensive survey." arXiv preprint arXiv:2207.07068 (2022).

- There are many more recent baselines for EO notion of fairness. For example, see the list below:

Mary, Jérémie, et al. "Fairness-aware learning for continuous attributes and treatments." ICML 2019.

Prost, Flavien, et al. "Toward a better trade-off between performance and fairness with kernel-based distribution matching." NeurIPS Workshops 2019.

Baharlouei, Sina, et al. "Rényi fair inference." ICLR 2020.

Cho, Jaewoong, et al. "A fair classifier using kernel density estimation." NeurIPS 2020.

Lowy, Andrew, et al. "A stochastic optimization framework for fair risk minimization." TMLR 2022.

- There is significant inconsistency between the notions of fairness considered in different parts of the paper. It is claimed in Section 1 that fairness is achieved if the accuracy is similar on different subgroups. This is *representation parity* notion of fairness, which is different from (and may even be incompatible with) the *equalized odds* notion that is considered subsequently.

- As it stands, no guarantees have been made about the generalization of fairness mitigation out-of-distribution. It would be great if the authors can better justify the choice of curvature for this problem (e.g., using theoretical arguments on toy problems).

- As far as I understand, the curvature of the loss landscape (which is the metric of interest for robustness in this paper) has been shown to be related to (adversarial) robustness in a small ball. This could be very different from natural distribution shifts with arbitrary changes in the distributions of interest.

- The comparisons as currently reported are not apples-to-apples. There is an inherent tradeoff between *fairness violation* and *accuracy*. Most SOTA methods, including all of the ones considered in this paper are capable of producing a Pareto front in the tradeoff curve between these two metrics by increasing the weight of the loss term associated with fairness. Therefore, the goal should be to produce a tradeoff curve whose Pareto front dominates the Pareto front of a different method. Currently the authors are reporting single points on this front which do not necessarily dominate each other. Hence, unfortunately the comparisons, as it stands, do not reveal much about the effectiveness of the method.

- *method*: The method's effectiveness has not been demonstrated on other fairness mitigation techniques other than adversarial training. As it stands, the applicability of the method seems narrow unless the authors can bring empirical evidence to the contrary.

- *fairness notions*: The method's effectiveness has not been demonstrated on other fairness metrics other than equalized odds. For example, the motivating factor in this paper was based on *representation parity.* Other popular notions are *demographic parity* and *equal opportunity*.

- *scaling*: The method's effectiveness has not been tested beyond binary classification with binary groups.

---

> ### Author Response · Authors · 2023-02-03
> **Response to Reviewer 4fit**
>
> Thank you for your insightful suggestions and careful reading. We have carefully addressed your concerns and suggestions below.
>
> 1. The cited literature in the paper is outdated.
>
> A: Thanks for pointing out the related works. We have added all of them in Section 2 using blue texts.
>
> 2. Inconsistency between the notions of fairness considered in different parts of the paper.
>
> A: Thank you for your careful reading. In this paper, we intended to focus on equalized odds, which overcomes the limitation of demographic parity [1]. We have updated Section 1 to align the definition of fairness to equalized odds using blue texts.
>
> [1] Equality of Opportunity in Supervised Learning.
>
> 3. Does the robustness achieved by curvature matching generalize to distribution shifts with arbitrary changes?
>
> A: Yes, you are right that the curvature of the loss landscape is related to adversarial robustness in a small ball, and such adversarial robustness leads to general OOD robustness as discussed in [2].
>
> [2] Improved OOD Generalization via Adversarial Training and Pre-training.
>
>
> 4. Replace tables with figures to show accuracy-fairness trade-off curves, and expand the scope of the effectiveness of the proposed method along the methods and fairness notion and scaling axes.
>
> A: Thanks for your suggestion. We have shown the trade-off curves in Figure 3. Following your suggestion, we extended Figure 3 by:
>
> 1) Adding a new  baseline method [3] based on distributionally robust training;
>
> 2) Adding equal opportunity as a new fairness metric.
>
> Following previous fairness learning works (Zhang et al., 2018; Madras et al., 2018), this paper focuses on the binary classification problem with binary privacy attributes. The proposed method can potentially be generalized to the multiclass classification or continuous prediction problems following similar ideas as in (Mary et al., 2019; Denis et al., 2021), which we leave as the future work. We added discussion on this limitation in Section 7 using blue texts.
>
> [3] Ensuring Fairness Beyond the Training Data. NeurIPS, 2020.
>
> 5. Details on MMD.
>
> A: As described in Section 4.3, we used a series of kernel widths: S = {1, 2, 4, 8, 16}, following previous works [4,5]. We use the whole batch of training samples to estimate MMD, where the batch size is 64. In other words, M+N=64 in Eq. (6). We have added this detail in Section 4.3 using blue texts.
>
> [4] Generative moment matching networks.
>
> [5] MMD GAN: Towards deeper understanding of moment matching network.
>
> 6. Add a broader impact section to discuss the limitations.
>
> Thank you for your suggestion. We added this as Section 7.
>
> 7. More inspection on the datasets.
>
> A: We added the “Oracal” method you suggested in the updated Figure 3, which is described in Section 5.1 using blue texts. Benchmarking on WILDS and Imagenet-C is a good suggestion. However, these datasets only have classification labels but don’t have privacy attribute annotations. Thus, our method and also previous methods like AdvDebias are not directly applicable on them without proper annotations on privacy attributes.
>
> 8. Other minor issues.
>
> A: Thanks for your careful reading and insightful comments. We have fixed all the minor issues you pointed out.

---

> > ### Comment · Reviewer_4fit · 2023-02-03
> > **Further comments.**
> >
> > Thanks for the revisions that address many of the issues that were previously raised. While I may have additional comments once I carefully read your response, here are some additional quick feedback that I hope you'd be able to address.
> >
> > - The reason I have been insisting on tradeoff curves is as follows. Consider a very simple baseline where the model predicts $\hat{y} = 0$ regardless of the features (or the sensitive attribute). By definition the model is both perfectly fair in-distribution and out-of-distribution and achieving $\Delta_{EO} = 0$ in both cases. Hence, it would be considered better than all baselines in Table 1. That is why reporting $\Delta_{EO}$ without regard to performance (e.g., model accuracy) doesn't make much sense.
> >
> > - I still think adding one or more experiments to consider equality of opportunity would add value to the paper to show the method extends beyond equalized odds.
> >
> > - Thinking more: I think your method might inherently improve adversarial robustness of the fairness mitigation techniques, and if true this could be a great selling point IMHO. It would be great if you can verify this. See (Mehrabi et al, 2021).
> >
> > Mehrabi, Ninareh, et al. "Exacerbating algorithmic bias through fairness attacks." Proceedings of the AAAI Conference on Artificial Intelligence. Vol. 35. No. 10. 2021.

---

> > > ### Author Response · Authors · 2023-02-08
> > > **Response to further comments**
> > >
> > > Thank you for providing the insightful further comments and detailed explanations. We agree that it is important to report the trade-off curves and add EOpp as another measurement. We have replaced the tables with figures showing trade-off curves and EOpp in the updated manuscript.
> > >
> > > Regarding the performance under adversarial attacks proposed in (Mehrabi et al, 2021), our empirical results show that our method does not bring significant benefit over the baseline methods against those adversarial attacks. Our method targets at improving robustness of fairness under "average-case" distributional shifts. Our assumption is that the "worst-case" adversarial attack in  (Mehrabi et al, 2021) is strong enough to break our method and the baseline methods, just like traditional adversarial attacks can break many robust training methods designed for "average-case" distributional shifts. We will leave it as the future work to study how to defend the attacks in (Mehrabi et al, 2021). We have added (Mehrabi et al, 2021) in the related works section. Thank you again for your insightful suggestion.

---

> > ### Comment · Reviewer_4fit · 2023-02-15
> > **Thanks for the revisions!**
> >
> > Thank you very much for the revisions, which I think have addressed my major comments. Here I list some further minor comments that need attention:
> >
> > - Please re-organize "Bias Mitigation Methods" on pages 3 and 4, which is currently very long. I suggest breaking it into several paragraphs (for example, discussing pre-processing, in-processing, and post-processing)
> >
> > - I think Mehrabi et al. doesn't belong to "Bias Mitigation Methods" and perhaps you can expand the scope of "Fairness under distributional shift" to more generally discuss robustness of fairness methods (both under distribution shift and adversarial attacks).
> >
> > - I think in "Evaluation Metrics" you may want to write a couple of sentences around why it is not meaningful to look at $\Delta_{EO}$ in isolation, as per our discussion in the other thread. It might be helpful to other researchers as well.
> >
> > - In Broader Impact, please try to also articulate what can go wrong with equalizing curvature. I think since we don't have a good theoretical understanding of the method, I would caution the reader on that end. I also suggest you add a few sentences around the negative result around robustness against adversarial attacks, which might be useful to a future reader.

---

### Comment · Action_Editors · 2023-01-19
**A message from AE**

Dear authors,

I would like to inform you that one more review will be coming in, and our EIC can extend the discussion period by a week or two. On the other hand, please don't wait for the last review and please begin your rebuttals just as our original timeline. Thanks for your understanding!

AE

---

> ### Comment · Reviewer_4fit · 2023-01-21
> **4th review is now submitted**
>
> Dear authors,
>
> The 4th review is now submitted. Thanks for your patience.
>
> Reviewer 4fit

---

### Decision · Action_Editors · 2023-02-18

**Recommendation:** Accept with minor revision

**Comment:**

This submission studied the robustness of fairness against distribution shifts. It demonstrated that the fairness achieved by previous methods (based on adversarial training) can no longer be guaranteed under even small distribution shifts. Then it advocated to use the loss curvature as the measure for robustness, and proposed a novel "curvature matching" solution following the idea that the loss curvatures of different data groups should be similar before and after a distribution shift happens. By this way, the proposed method can successfully transfer the fairness obtained on the training distribution to the test distribution.

The submission received 4 reviews from 4 experts in different research areas --- fairness, robustness, distribution shift, and domain adaptation. There were some concerns about the experiment design, and fortunately the authors addressed the issues. In the end, all 4 reviewers voted for acceptance, and thus we should definitely accept the paper for publication! Note that there are "some further minor comments that need attention" from Reviewer 4fit (who like the paper most among the reviewers), so please take them into account (if possible) in the final version.

**Audience:**

Yes, all 4 reviewers agreed that the studied problem is extremely important to the community.

**Claims And Evidence:**

Yes.